# Change Trajectory of Symptom Distress, Coping Strategies, and Spiritual Wellbeing in Colorectal Cancer Patients Undergoing Chemotherapy

**DOI:** 10.3390/healthcare11060857

**Published:** 2023-03-14

**Authors:** Ching-Wen Wei, Shu-Yuan Liang, Chia-Hui Chin, Hua-Ching Lin, John Rosenberg

**Affiliations:** 1Department of Nursing, Tzu Chi University of Science and Technology, 880, Section 2, Chien-Kou Road, Hualien 970, Taiwan; bb831114mm@gmail.com; 2School of Nursing, National Taipei University of Nursing and Health Sciences, 365 Ming Te Road, Beitou, Taipei 112, Taiwan; 3Department of Nursing, Shin Kong Wu Ho-Su Memorial Hospital, 95 Wen Chan Road, Shih Lin, Taipei 112, Taiwan; 4Division of Colorectal Surgery, Chen-Hsin General Hospital, Taipei 112, Taiwan; linhuaching3@gmail.com; 5School of Health, University of the Sunshine Coast, Queensland, Australia 80-106 Tallon Street, Caboolture, QLD 4059, Australia; jrosenbe@usc.edu.au

**Keywords:** cancer, coping strategy, symptom distress, spiritual wellbeing, trajectory

## Abstract

Impacts caused by cancer and associated treatment may change with time. The objective of this study is to examine the change trajectory of symptom distress, coping strategies, and spiritual wellbeing in colorectal cancer patients during chemotherapy and to further examine the predictors of spiritual wellbeing. A prospective longitudinal repeated measures study design was employed. A total of 97 patients undergoing chemotherapy for the first time were enrolled. A structured questionnaire was used to collect data at three timepoints, which were before chemotherapy (T0), during chemotherapy (T1: 3 months after T0), and after chemotherapy (T2: 6 months after T0). The results of this study show that patients have significantly lower spiritual wellbeing and significantly higher symptom distress during chemotherapy treatment (T1). Family support (*B* = 0.39, *p* = 0.007) and problem-focused coping strategies (*B* = 0.47, *p* = 0.001) are significant predictors of spiritual wellbeing before chemotherapy (T0). Symptom distress (*B* = −0.18, *p* = 0.048) and problem-focused coping strategies (*B* = 0.26, *p* = 0.028) are significant predictors of spiritual wellbeing during chemotherapy (T1). The results provide care recommendations for different stages of chemotherapy to help to achieve more precise patient care and improve care quality.

## 1. Introduction

New colorectal cancer patients account for 9.5% of new cancer patients globally each year, ranking third after breast cancer and lung cancer in new cancer cases globally [1,2]. According to the Taiwan Ministry of Health and Welfare, 42.9 per 100,000 people develop colorectal cancer every year, and the mortality rate increment is 0.9%. Colorectal cancer is the third leading cause of death among the top ten causes of cancer death and tends to occur in elderly people aged 50 to 79 years [3]. In order to improve patient survival rate and decrease the risk of cancer metastases, surgery, chemotherapy, and radiotherapy are combined as the main treatment method. However, the side effects caused by treatment impact the overall health status of colorectal cancer patients and persistently affect their spiritual wellbeing during disease progression [4,5].

According to the definition by Hungelmann et al. [6,7] spiritual wellbeing is ‘‘A sense of harmonious interconnectedness between self, others/nature, and Ultimate Other which exists throughout and beyond time and space. It is achieved through a dynamic and integrative growth process which leads to a realization of the ultimate purpose and meaning of life’’. Spiritual wellbeing is an important factor for maintaining physical and mental health, as it helps an individual in integrating and improving the meaning of life, as well as facing challenges caused by the disease and associated treatment. Spiritual wellbeing is a type of care practice utilized during disease progression and cancer treatment [5]. Related studies also show that spiritual wellbeing is an important care issue during disease progression in colorectal cancer [8], particularly when faced with symptom distress caused by chemotherapy [9,10,11]. However, spiritual wellbeing tends to be overlooked, as it can be conceptually abstract, difficult to evaluate, and challenging to implement due to medical staff being busy [10].

Symptom distress is a common problem encountered by patients during cancer treatment and may include physiological and psychological distress [12]. The type and intensity of symptom distress experienced by patients during cancer treatment may differ due to the number of chemotherapy cycles. For example, before undergoing chemotherapy, and during the early stages of chemotherapy, patients may have greater psychological symptom distress due to fear and worry over diagnosis and treatment [12]. Subsequently, with the onset of chemotherapy side effects, patients may experience more physiological symptom distress.

Studies show that symptom distress progression depicts a decreasing trend with time after the middle of treatment [13]. In approximately 20% to 38% of colorectal cancer patients, symptom distress caused by cancer treatment will continue after conclusion of treatment [14], and symptom distress significantly affects the spiritual wellbeing of patients [9]. However, coping strategies employed by patients can help their spiritual wellbeing [15].

Lazarus et al. [16] defined coping as “the constant cognitive change and behavioral adaptation when handling specific external and/or internal demands that are evaluated as something that exceeds the resources of the person”. Coping is a dynamic process, which is a process of interaction between the individual and the environment to produce a series of influences to each other. Coping includes elements, such as cognition, behavior, and strategies, to cope with the negative impact of stressful events.

In the face of stress caused by cancer and its treatment, patients will evaluate their internal or external resources and adopt suitable coping strategies, such as problem-focused coping, emotion-focused coping, and meaning-focused coping [17,18] based on the evaluation results [19]. Studies show that coping strategies employed by patients are significantly correlated with their spiritual wellbeing [15]. Therefore, when faced with impacts due to cancer and its treatment, individual coping strategies also play an important buffering role.

Faced with various impacts due to cancer and its treatment, the severity of its effect may change due to treatment duration and disease progression. However, there are very few studies which have carried out long-term follow-up on the changes in symptom distress, coping strategies, and spiritual wellbeing in colorectal cancer patients during chemotherapy. In order to understand the health status of colorectal cancer patients from diagnosis to the end of chemotherapy, this study examined the correlation between symptom distress, coping strategies, and spiritual wellbeing and their trajectory changes in colorectal cancer patients; and, furthermore, the predictors of spiritual wellbeing were examined.

## 2. Methods

### 2.1. Study Design and Subjects

A prospective longitudinal repeated measures study design was employed. Convenience sampling was used to enroll patients from a colorectal department ward in a district hospital in northern Taiwan. The enrollment period was from October 2019 to December 2021. There were three data collection timepoints, which were before the first chemotherapy cycle (T0), in the middle of chemotherapy (T1: 3 months after T0), and after chemotherapy (T2: 6 months after T0). G-power 3.1.3 was used to calculate the sample size and a F test multiple linear regression model: a fixed model, representing R^2^ deviation from zero, was used as a basis. The effect size was set to 0.2, and the significance level was set to 0.05, and the power was set to 0.85. As the enrollment period lasted for six months, the expected loss to follow-up rate was set to 20%. The estimated sample size was 92 patients, and 97 patients were ultimately enrolled. The inclusion criteria included: (1) new patients with a definite diagnosis of colorectal cancer; (2) chemotherapy-naive; (3) clear consciousness; (4) able to communicate in Chinese and Taiwanese Hokkien; and (5) aged 20 years and above. The exclusion criteria included patients with psychiatric disorders, depression, and/or anxiety.

### 2.2. Study Process

Prior to enrollment, approval was first obtained from the institutional review board (IRB No: (722) 108A-38). The subject was informed of the study and provided an informed consent form with content, including the study objectives, enrollment period, study process, and investigator contact information. Subjects were informed that participation is voluntary, and they can request to stop or withdraw from the study at any time without a reason and this will not affect their treatment or care rights. The patient was enrolled after they expressed their understanding and signed the informed consent form. The investigator provided all of the questionnaires for the patient to complete themselves. If the patients were unable to answer the questionnaire due to cognitive ability. The investigator read out the contents of the questionnaire items and then asked the patient to answer the options. The investigator filled in the options answered by the patients. After the patient had completed the form, the investigator inspected the completed form onsite to confirm if there were any omissions. The medical characteristics of the patients were collected from their medical records by the investigator.

### 2.3. Study Tools

The general demographic characteristics and medical variables of the patients include gender, age, marital status, education level, religion, family income, family support, employment, history of chronic disease, smoking, alcohol consumption, cancer stage, and type of chemotherapy. The structured questionnaire includes the Functional Assessment of Chronic Illness Therapy-Spiritual Well-being (FACIT-Sp-12), which was used to measure the spiritual wellbeing of patients. This was the spiritual wellbeing subscale in Functional Assessment of Cancer Therapy—General (FACT-G). The FACT-G scale was originally developed to evaluate the quality of life of cancer patients [20]. This scale uses the 5-point Likert scale, of which 0 means not at all, and 4 means very much. There are 12 questions in total, and the total score ranges from 0–48. The higher the score, the better the spiritual wellbeing. The 12 questions are divided into two dimensions (meaning/peace and faith), with eight questions in the meaning/peace component and four questions in the faith component. Two questions were negatively worded in meaning/peace dimension. The alpha coefficients of the FACIT-Sp-12 scale and its subscales ranged from 0.81 to 0.88 and was positively correlated with FACT-G (*r* = 0.58) and negatively correlated with total mood disturbance score (*r* = −0.54) [20]. Bovero et al. [15] previously employed FACIT-Sp-12 to examine spiritual wellbeing in cancer patients and revealed that the two subscales of FACIT-Sp-12 and Hospital Anxiety and Depression Scale show significant divergent validity (*r* = −0.566 and *r* = −0.622).

#### 2.3.1. M. D. Anderson Symptom Inventory-Taiwan Form (MDASI-T)

MDASI-T was used to measure symptom distress in patients. The original scale was developed by the University of Texas M.D. Anderson Cancer Center and translated into the Taiwanese Chinese version by Lin et al. [21]. This scale includes 13 common symptoms experienced in cancer patients (i.e., pain, fatigue, nausea, sleep disorders, depression, polypnea, forgetfulness, loss of appetite, drowsiness, dry mouth, psychological pain, vomiting, feeling of numbness) and six daily functional interferences caused by symptoms (i.e., general activities, mood, work, interpersonal relations, walking ability, enjoyment of life). This scale uses an 11-point Likert scale ranging from 0–10, in which 0 indicates no, and 10 indicates the highest severity of a symptom or interference with daily functions. The scale includes two subscales of symptom severity and symptom interference, and their Cronbach’s α values were 0.89 and 0.94, respectively [21]. This scale was previously used to examine the symptom distress of colorectal cancer patients and bladder cancer patients and its Cronbach’s α ranged from 0.84 to 0.94 [22,23].

#### 2.3.2. Jalowiec Coping Scale (JCS)

JCS was used to measure coping strategies in patients [24,25] and Chiou [26] translated and revised the JCS into the Chinese coping scale. The entire scale contains 40 questions, including two subscales. There are 15 questions in the problem-focus coping subscale and 25 questions in the emotion-focus coping subscale. The scale uses a 5-point Likert scale, in which 0 means never used, and 4 mean often used. The higher the subscale scores, the higher the coping strategy frequency used by the patients. The Cronbach’s α of the original total scale and subscales ranges from 0.79 to 0.86. The Cronbach’s α of the Chinese total scale is 0.78 [26].

### 2.4. Data Analysis

The SPSS/PC 20.0™ software was used in this study for coding, filing, and data analysis. Descriptive statistics include frequency, percentage, mean, and standard deviation (SD), and they were used to present the demographics, medical characteristics, symptom distress, coping strategies, and spiritual wellbeing scores at different timepoints in colorectal cancer patients. An independent t-test, one-way ANOVA, repeated measured ANOVA, Pearson’s correlation, and multiple regression models were used for inferential statistics to examine the correlation between variables, differences between different timepoints, and important predictors. For repeated measure, ANOVA and LSD were used for post hoc comparison of results. A difference of *p* < 0.05 was considered to be statistically significant.

## 3. Results

### 3.1. Relationship between Patient Demographics, Medical Characteristics, and Spiritual Wellbeing

A total of 97 patients were enrolled in this study. The mean age was 61.6 years (SD = 10.0). Most patients were older than 66 years (40.2%), male (56.7%), married (68.0%), religious (84.5%), with a family monthly income less than 3500 USD (73.2%), having good perceived family support (72.2%), and were unemployed (77.3%). Thirty-eight percent were senior high school and vocational school graduates. Out of the sample, 45.4% of patients had other chronic diseases, and 45.4% of patients had stage III cancer (Table 1).

In this study, a t-test and one-way ANOVA were used to test the differences in spiritual wellbeing between demographic and medical characteristics in patients. Results showed that only family support depicted significant differences in spiritual wellbeing (*t* = −2.89, *p* = 0.005) (Table 1).

### 3.2. Scores and Trends of Symptom Distress, Coping Strategies and Spiritual Wellbeing

The overall symptom distress score of patients was the highest in T1 (M = 2.42, SD = 1.79), which was significantly higher than before chemotherapy (T0) (M = 1.68, SD = 1.11), but it showed a decreasing trend after chemotherapy (T2) (M = 1.78, SD = 1.41). Additionally, there were significant differences between T0 and T1 (*F =* 16.28, *p* < 0.001).

Overall coping strategies and their components were the highest at T0. For overall coping strategies, the T0 score was the highest (M = 2.51, SD = 0.47), and the T1 score was the lowest (M = 2.38, SD = 0.37). There were statistically significant differences in overall coping strategies (*F =* 4.82, *p* < 0.05) and problem-focused coping strategies (*F =* 5.40, *p* < 0.01) between T0 and T1.

Overall spiritual wellbeing and its components were the highest at T0. For overall spiritual wellbeing, the T0 score was the highest (M = 32.82, SD = 7.16), and the T1 score was the lowest (M = 31.08, SD = 6.31). There were statistically significant differences in overall spiritual wellbeing (*F =* 4.31, *p* < 0.05) and meaning/peace (*F =* 5.77, *p* < 0.01) between T0 and T1, but there was no statistically significant differences in faith between T0 and T1 (*F =* 1.78, *p* > 0.05) (Table 2 and Figure 1).

### 3.3. Correlation between Symptom Distress, Coping Strategies, and Spiritual Wellbeing

Pearson’s correlation was used to analyze the correlation between symptom distress, coping strategies, and spiritual wellbeing. The results showed that there was no significant correlation between symptom distress and spiritual wellbeing (*r* = 0.036, *p* > 0.05), problem-focused coping strategy showed a significant and positive correlation with spiritual wellbeing (*r* = 0.439, *p* < 0.01), and emotion-focused coping was not significantly correlated with spiritual wellbeing (*r* = −0.160, *p* > 0.05) (Table 3).

### 3.4. Predictors of Spiritual Wellbeing

A multiple regression was used to examine the predictors of spiritual wellbeing in patients. Among all demographic variables, family support showed significant differences in patient spiritual wellbeing (*t* = −2.89, *p* = 0.005) (Table 1). Therefore, this variable was included in the statistical analysis of predictive variables for spiritual wellbeing. Family support, symptom distress, and coping strategies were included in the multiple regression model to identify important predictors of spiritual wellbeing before chemotherapy (T0), in the middle of chemotherapy (T1), and after treatment (T2) in colorectal cancer patients. The results showed that family support (*B* = 0.39, *p* = 0.007) and problem-focused coping strategy (*B* = 0.47, *p* = 0.001) could significantly predict spiritual wellbeing before chemotherapy (T0). Compared to patients with fair family support, patients with good family support had 0.39 more spiritual wellbeing points. Every 1 point increase in problem-focused coping strategy increased spiritual wellbeing by 0.47 points. This model has 52% explanatory power (*F =* 8.60, *p* = 0.001). In the middle of chemotherapy (T1), every 1 point increase in symptom distress decreases spiritual wellbeing by 0.18 points (*B* = −0.18, *p* = 0.048). Every 1 point increase in problem-focused coping strategy increased spiritual wellbeing by 0.26 points (*B* = 0.26, *p* = 0.028). However, family support, symptom distress, and coping strategies are not predictors of spiritual wellbeing after chemotherapy (T2) (Table 4).

## 4. Discussion

The results show that spiritual wellbeing and coping strategies were the best before chemotherapy (T0) and were the worst in the middle of chemotherapy (T1, three months after T0). In comparison, symptom distress was the worst in the middle of chemotherapy (T1) and decreased after chemotherapy (T2, six months after T0). Family support and problem-focused coping strategy significantly predict spiritual wellbeing before chemotherapy (T0). In the middle of chemotherapy (T1), symptom distress and problem-focused coping strategy significantly predict spiritual wellbeing. At the end of chemotherapy (T2), family support, symptom distress, and coping strategies do not predict spiritual wellbeing.

The results of this study show that, among symptom distress, coping strategies, and spiritual wellbeing, the magnitude of change with treatment course was the greatest for symptom distress. Although symptom interference in symptom distress at the end of chemotherapy (T2) was lower than before chemotherapy (T0) and in the middle of chemotherapy (T1), symptom severity in the middle of chemotherapy (T1) and end of chemotherapy (T2) were significantly higher than before chemotherapy (T0). Overall, symptom distress in the middle of chemotherapy (T1) was significantly higher than before chemotherapy (T0). The results of this study support the fact that chemotherapy causes symptom distress. Although symptom distress slightly decreased at the end of chemotherapy (T2), symptom distress causes physiological and psychologic impact.

Related studies show that symptom distress was the most severe after the first chemotherapy cycle and that its severity gradually decreases with the number of chemotherapy cycles and duration [27,28]. However, a two-year long-term follow-up study by Qaderi et al. [28] found that, although symptom distress in colorectal cancer patients decreases with time, it may extend after conclusion of treatment.

This study showed that coping strategies were the best before chemotherapy (T0) and the worst in the middle of chemotherapy (T1). Even at the end of chemotherapy, coping strategies did not return to pre-chemotherapy status, and the trends for problem-focused and emotion-focused coping strategies showed similar trends. On the other hand, in this study, the frequency of using problem-focused coping in patients was greater than the frequency of emotion-focused coping, which was similar to the results of Sari et al. [29], who found that 65.7% of cancer patients employ problem-focused coping during chemotherapy, and only 30.4% employ emotional-focused coping [29]. Furthermore, some studies also support that different cancer patients will employ different strategies when initially faced with treatment [30]. For example, fewer colorectal cancer patients tend to employ avoidance coping or emotional coping compared with breast cancer patients. However, the frequency of using emotion-focused coping for patients in the current study changes as symptom distress severity increases. More specifically, this might mean that patients face resource utilization or symptom management challenges when faced with symptom distress [18].

This study shows that problem-focused coping strategy is an important predictor of spiritual wellbeing before chemotherapy (T0) and in the middle of chemotherapy (T1). Problem-focused coping strategy is a coping strategy of active problem analysis, seeking resources, and carrying out actions, while the emotion-focused coping strategy is more focused on changing one’s thoughts when an individual is powerless to change the current environment or stress [18]. Therefore, professional medical staff should strengthen problem solving in colorectal cancer patients undergoing chemotherapy, particularly when faced with symptom distress.

This study shows that family support is the only sociodemographic variable that is significantly related to patient spiritual wellbeing. Family support can include informational support, material/instrumental support, physical support, and emotional support [31]. With advances in medical techniques, cancer patient treatment has gradually transitioned to outpatient and home treatment. Therefore, family support plays an important role in home self-care of cancer patients during treatment or disease recovery stages [32,33].

The study by Sari et al. [29] showed that patients with good family support have 12.16 times better problem-focused coping than those with poor family support. On the other hand, patients with good family support have 0.14 times better emotion-focused coping than cancer patients with poor family support (OR < 1) [29]. Hence, patients with good family support have better mental health [33] and self-esteem [34]. In addition, family support also shows a significant positive correlation with spiritual wellbeing in patients [35]. Family support can allow cancer patients to feel love and concern and the support process can help in self-understanding in cancer patients and accept disease as a part of life [34]. This may affect physical and mental recovery, as well as spiritual health, in cancer patients.

This study found that, when colorectal cancer patients enter the middle of chemotherapy (T1), overall spiritual wellbeing, the meaning/peace component, and the faith component all became worse. At the same time, patients had higher symptom distress. In addition, symptom distress significantly predicts spiritual wellbeing in this stage (T1). Related studies also support that cancer patients with low symptom distress during chemotherapy have better spiritual wellbeing [36,37]. Kamijo and Miyamura [36] examined spiritual wellbeing in cancer patients during chemotherapy. The results showed that related symptoms (such as poor appetite or insomnia) will decrease spiritual wellbeing in patients [36]. Compared with the changes in the trajectory of symptom distress, the trajectory of spiritual well-being in the current study is relatively stable. Other studies related to spiritual wellbeing have a trajectory similar to the results of the current study [38,39]. Results in a population of brain malignancies show that spiritual wellbeing is relatively stable along the course of the cancer [38].

Spiritual wellbeing is an indispensable factor in personal health. The results of this study showed that the predictors of spiritual wellbeing will change with time. Family support and problem-focused coping are important predictors of spiritual wellbeing before chemotherapy (T0). In the middle of chemotherapy (T1), symptom distress caused by treatment is a significant predictor of spiritual wellbeing. At this time, family support is no longer a significant predictor of spiritual wellbeing. However, at this time, problem-focused coping is still a significant predictor of spiritual wellbeing. At the end of chemotherapy (T2), family support, symptom distress, and coping strategies are no longer significant predictors of spiritual wellbeing in patients. Besides reflecting the resolution of symptom distress, this reflects that the coping strategies of patients in this study did not recover to before chemotherapy (T0) at the end of chemotherapy (T2).

In summary, the results of this study showed that colorectal cancer patients have significantly lower spiritual wellbeing and significantly higher symptom distress in the middle of chemotherapy (T1). Problem-focused coping strategies are significant predictors of spiritual wellbeing before chemotherapy (T0) and in the middle of chemotherapy(T1). Family support is only a significant predictor of spiritual wellbeing before chemotherapy (T0), and symptom distress is only a significant predictor of spiritual wellbeing in the middle of chemotherapy (T1).

### Study Limitations

This study only included patients in a hospital in northern Taiwan, and most enrolled patients were middle-aged to elderly patients. The follow-up period was only six months (i.e., early stage of chemotherapy completion), which may affect the inference of the study results.

## 5. Conclusions

Based on the results of this study, family support resources should be evaluated, and coping strategies should be strengthened in patients before chemotherapy, particularly problem-focused coping strategies. During the middle of chemotherapy, management of symptom distress should be strengthened in patients. The results of the current longitudinal study were based on the trajectory changes in physical and mental impacts of chemotherapy on colorectal cancer patients. The results provide care recommendations for different stages of chemotherapy and help to achieve more precise care and improve care quality for patient with colorectal cancer.

## Figures and Tables

**Figure 1 healthcare-11-00857-f001:**
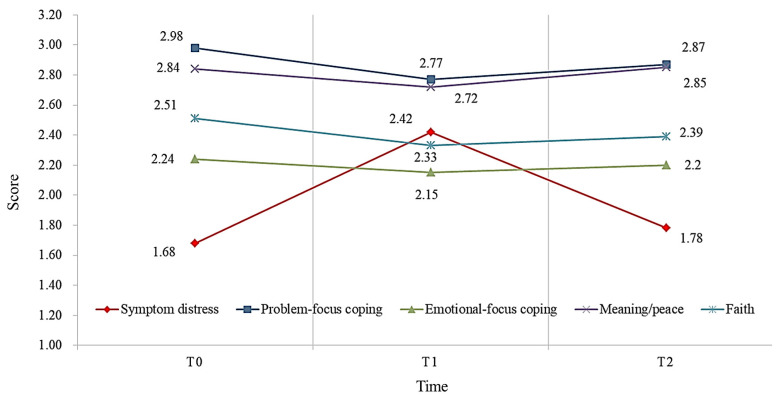
Trends of symptom distress, coping strategies, and spiritual wellbeing trends (N = 97).

**Table 1 healthcare-11-00857-t001:** Demographic and medical characteristics by spiritual wellbeing (N = 97).

Variable	n (%)		Spiritual Wellbeing	*t/F*	*p*-Value
			M ± SD		
Sex				0.18	0.98
Male	55	56.7	32.84 ± 7.30		
Female	42	43.3	32.81 ± 7.07		
Age (years)				0.86	0.42
≤55	29	29.9	33.76 ± 6.49		
55-65	29	29.9	33.45 ± 7.14		
≧66	39	40.2	31.67 ± 7.67		
Marital status				0.40	0.68
Unmarried/divorced/widowed	31	32.0	33.26 ± 6.63		
Married	66	68.0	32.62 ± 7.44		
Education level				2.59	0.08
Junior high school and below	33	34.0	30.88 ± 7.04		
Senior high school and vocational school	37	38.1	32.95 ± 7.43		
University and above	27	27.8	35.04 ± 6.48		
Religion				−0.32	.74
No	15	15.5	32.27 ± 6.53		
Yes	82	84.5	32.93 ± 7.30		
Family monthly income (USD)				0.03	0.96
≤3500	71	73.2	32.72 ± 7.31		
3501–7100	19	19.6	33.00 ± 5.94		
≧7101	July	7.2	33.43 ± 9.53		
Family support				−2.89	0.005
Fair	27	27.8	29.56 ± 6.69		
Good	70	72.2	34.09 ± 6.68		
Employment				−1.38	0.16
No	75	77.3	32.28 ± 7.31		
Yes	22	22.7	34.68 ± 6.42		
History of chronic disease				−1.62	0.10
No	53	54.6	31.75 ± 6.25		
Yes	44	45.4	34.11 ± 8.01		
Smoking				−0.17	0.86
No	71	73.2	32.75 ± 7.26		
Yes	26	26.8	33.04 ± 7.02		
Alcohol consumption				−1.08	0.28
No	80	82.5	32.46 ± 7.29		
Yes	17	17.5	34.53 ± 6.44		
Cancer stage				0.16	0.85
II	32	33.0	33.41 ± 7.03		
III	44	45.4	32.61 ± 7.43		
IV	21	21.6	32.38 ± 7.06		
Type of chemotherapy				1.17	0.31
Oxalip	56	57.7	33.34 ± 6.96		
Campto	37	38.1	31.65 ± 7.65		
Oral	4	4.1	36.50 ± 3.31		

**Table 2 healthcare-11-00857-t002:** Difference in symptom distress, coping strategies, and spiritual wellbeing scores between timepoints (N = 97).

Variables	T0	T1	T2	*F*	*p*-Value	LSD
	M ± SD	M ± SD	M ± SD			
Symptom distress	1.68 ± 1.11	2.42 ± 1.79	1.78 ± 1.41	16.28	<0.001	T1 > T0
Severity	2.00 ± 1.31	3.32 ± 1.56	2.52 ± 1.93	23.36	<0.001	T1 > T2 > T0
Interference	1.36 ± 1.12	1.53 ± 1.02	1.04 ± 0.98	9.14	<0.001	T0, T1 > T2
Coping strategies	2.51 ± 0.47	2.38 ± 0.37	2.44 ± 0.42	4.82	0.012	T0 > T1
Problem-focus	2.98 ± 0.69	2.77 ± 0.60	2.87 ± 0.68	5.40	0.008	T0 > T1
Emotional-focus	2.24 ± 0.46	2.15 ± 0.35	2.20 ± 0.37	2.16	0.126	
Spiritual wellbeing	32.82 ± 7.16	31.08 ± 6.31	32.44 ± 6.04	4.31	0.017	T0, T2 > T1
Meaning/peace	22.76 ± 4.18	21.75 ± 3.74	22.86 ± 3.32	5.77	0.005	T0, T2 > T1
Faith	10.06 ± 4.07	9.33 ± 3.47	9.59 ± 3.53	1.78	0.171	

Note: T0: before the first chemotherapy cycle; T1: in the middle of chemotherapy; T2: after the last chemotherapy cycle.

**Table 3 healthcare-11-00857-t003:** Correlation between symptom distress, coping strategies, and spiritual wellbeing (N = 97).

Variables		Symptom Distress	Coping Strategies	
			Problem	Emotional
Symptom distress		1		
Coping strategies	Problem	0.127	1	
	Emotional	−0.289	0.072	1
Spiritual wellbeing	Meaning/peace	−0.075	0.504 **	−0.098
	Faith Total scale	0.141 0.036	0.566 ** 0.439 **	−0.030 −0.160

** *p* < 0.01.

**Table 4 healthcare-11-00857-t004:** Predictors of spiritual wellbeing (N = 97).

Variables	T0			T1			T2		
	*B*	*t*	*p*-Value	*B*	*t*	*p*-Value	*B*	*t*	*p*-Value
Family support Fair (ref) vs. good	0.39	2.74	0.007 **	0.13	1.33	0.184	0.20	1.77	0.079
Symptom distress	−0.20	−0.33	0.742	−0.18	−1.74	0.048 *	−0.07	−0.26	0.797
Problem coping	0.47	4.87	0.001 **	0.26	2.23	0.028 *	0.11	1.26	0.209
Emotion coping	−0.17	−1.91	0.060	−0.11	−0.97	0.330	0.02	0.25	0.799
*R^2^*	0.52			0.30			0.25		
*F*	8.60			2.23			1.64		
*p*-value	0.001 **			0.045 *			0.169		

Note: * *p* < 0.05, ** *p* < 0.01; T0: before the first chemotherapy cycle; T1: in the middle of chemotherapy; T2: after the last chemotherapy cycle.

## Data Availability

The data presented in this study are available upon request from the corresponding author upon reasonable request.

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
