# Peer review of "Change Trajectory of Symptom Distress, Coping Strategies, and Spiritual Wellbeing in Colorectal Cancer Patients Undergoing Chemotherapy"

_healthcare, 2023, doi:10.3390/healthcare11060857_

Round 1

Reviewer 1 Report

Ching-Wen Wei et al examine the change trajectory of symptom distress, coping strategies, and spiritual wellbeing in colorectal cancer patients during chemotherapy and to examine the predictors of spiritual wellbeing. They found that Family support and problem-focused coping strategies are predictors of spiritual wellbeing before chemotherapy. Symptom distress and problem-focused coping strategies are predictors of spiritual wellbeing during chemotherapy.

In my opinion, the topic is interesting to the clinical doctors and nurses, and the results are helpful to guide the nursing for the patients recovery after treatment.

However, there were several major concerns need to be addressed:

1. The present of results is bad and need to revised according to the recognized format.

2. The data in the results part should present as mean±SD, please do not separate it.

3. In Table 1, there was mistake at the row of “Age (years) (M ±SD)”

4. In page 5, first paragraph of Results 3.2, this content is description of statistics, they should not present.

5. In the first paragraph of Discussion, “The objective of this study is to examine…..” this sentence was inappropriate placed, it should be place at the Introduction part.

6. The conclusion is too long, please used concise sentence to describe.

Author Response

Review 1

Comments and Suggestions for Authors

Ching-Wen Wei et al examine the change trajectory of symptom distress, coping strategies, and spiritual wellbeing in colorectal cancer patients during chemotherapy and to examine the predictors of spiritual wellbeing. They found that Family support and problem-focused coping strategies are predictors of spiritual wellbeing before chemotherapy. Symptom distress and problem-focused coping strategies are predictors of spiritual wellbeing during chemotherapy.

In my opinion, the topic is interesting to the clinical doctors and nurses, and the results are helpful to guide the nursing for the patients recovery after treatment.

However, there were several major concerns need to be addressed:

  1. The present of results is bad and need to revised according to the recognized format.

Response: We revise present of results according the following suggestions.

  1. The data in the results part should present as mean±SD, please do not separate it.

Response: We revise to mean±SD or M±SD.

  1. In Table 1, there was mistake at the row of “Age (years) (M ±SD)”

Response: We delete M±SD of age in Table 1. It had put in the text of Results.

  1. In page 5, first paragraph of Results 3.2, this content is description of statistics, they should not present.

Response: We delete the description of statistics in the first paragraph of Results 3.2.

  1. In the first paragraph of Discussion, “The objective of this study is to examine…..” this sentence was inappropriate placed, it should be place at the Introduction part.

Response: We delete this sentence placed in the Discussion.

  1. The conclusion is too long, please used concise sentence to describe.

Response: We do some reshaping to shorten the conclusion.

Reviewer 2 Report

In the introduction, please cite the definition of spirituality according to two independent authors.

Under Methods, write whether the respondents had trouble filling out the survey questionnaire. If they had trouble filling it out, how did the authors of the survey help them fill it out.

Please explain why the study group consisted of 97 patients - this is quite a small study group. I do not know whether to designate such a small study group, that these are preliminary tests for further, much larger population studies.

Please, please, more literature, specifically literature.

Discussion first in the summary section of the study results. Please edit and not only the study of spirituality in the aspect of colorectal cancer but also other cancers or chronic diseases.

Author Response

Response to Reviewer 2 Comments

Review 2

In the introduction, please cite the definition of spirituality according to two independent authors.
Response: We add the definition of spiritual wellbeing in the Introduction.

Under Methods, write whether the respondents had trouble filling out the survey questionnaire. If they had trouble filling it out, how did the authors of the survey help them fill it out.
Response: We put more description of the investigator done for the respondents had trouble filling out the survey questionnaire.

Please explain why the study group consisted of 97 patients - this is quite a small study group. I do not know whether to designate such a small study group, that these are preliminary tests for further, much larger population studies.
Response: The study group consisted of 97 patients is according to G-power analysis. G-power 3.1.3 was used to calculate the sample size and a F test multiple linear regression model: Fixed model, R2 deviation from zero was used as a basis. The effect size was set to 0.2, the significance level was set to 0.05, and the power was set to 0.85. As the enrollment period lasted for six months, the expected loss to follow-up rate was set to 20%. We described the G-power analysis in the study design and subjects in the Methods.  

Please, please, more literature, specifically literature.
Response: We add more literatures in the Introduction including the definition of spiritual wellbeing and coping.

Discussion first in the summary section of the study results. Please edit and not only the study of spirituality in the aspect of colorectal cancer but also other cancers or chronic diseases.

Response: Yes, we do some revisions to show the discussion first in the summary section of the study results and add more other cancers and population to show the spiritual trajectory.

Reviewer 3 Report

Thanks for the authors. A very good and interesting study was conducted. 

The objective of this study is to examine the change trajectory of symptom distress, coping strategies, and spiritual  wellbeing in colorectal cancer patients during chemotherapy and to further examine the predictors  of spiritual wellbeing. A prospective longitudinal repeated measures study design was employed.  Very good!But there are some issues that need to be solved by the author.  

1.  What is the evidence for selecting the time points in this study?  Why not based on the number of chemotherapy cyclesHow many times do your patients undergo chemotherapy in total ?  

2.  Lines 39-40,   please check,  the third   or   second?  

3.  Line 92     R2  should be R2  

4.  Are there any cases lost at three time points?    

5.  Line 163     p< .05    should  be     p≤ .05  

6.  please  improve  table 3,   supplement  the   Correlation between symptom distress, coping strategies, and spiritual wellbeing and its subdomains

7.  Table  4 ,Is  R2     adjusted R???     please   supplement B value   for  every variable.   

8.  Lines  223-229,I suggest using B values to explain these results.

9.   Please  change the  references format   according  to   MDPI   journals.

 Best regards  and  good  luck

Author Response

Response to Reviewer 3 Comments

Review 3

Thanks for the authors. A very good and interesting study was conducted. 

The objective of this study is to examine the change trajectory of symptom distress, coping strategies, and spiritual  wellbeing in colorectal cancer patients during chemotherapy and to further examine the predictors  of spiritual wellbeing. A prospective longitudinal repeated measures study design was employed.  Very good!But there are some issues that need to be solved by the author.  

  1. What is the evidence for selecting the time points in this study?  Why not based on the number of chemotherapy cycles?How many times do your patients undergo chemotherapy in total? 

Response: At present, chemotherapy for colorectal cancer is performed once every two weeks, a total of 12 times, and it takes about 6 months to complete. Although some studies have shown that when breast cancer patients receive chemotherapy, the symptom distress will be the most serious in the first month after chemotherapy, but for colorectal cancer, the current clinical practice has found that about 80% of patients with more serious symptoms appeared when after the 5th to 7th chemotherapy (about the 3rd month), so we set data collection before chemotherapy, 3 months after chemotherapy, and after completion of chemotherapy.

2.Lines 39-40,   please check,  the third   or   second?

Response: We check it again, Colorectal cancer is the third leading cause of death among the top ten causes of cancer death in Taiwan.

  1. Line 92     R2  should be R2

Response: We revise it to R2.

  1. Are there any cases lost at three time points?

Response: All the patients that entered the study at the beginning completed the questionnaires at three time points.

5.Line 163     p< .05    should  be     p≤ .05 

Response: Thank you, however we used p < .05 based on the advices of statistician.

  1. please  improve  table 3,  supplement  the   Correlation between symptom distress, coping strategies, and spiritual wellbeing and its subdomains

Response: We revise table 3 with supplement the correlation between symptom distress, coping strategies, and spiritual wellbeing and its subdomains.

  1. Table  4 ,Is  R2    adjusted R2  ???     please   supplement Bvalue   for  every variable.

Response: We check the data again, it is R2. We supply B value for every variable in Table 4.

  1. Lines  223-229,I suggest using Bvalues to explain these results.

Response: We revise to explain these results use B values.

  1. Please  change the  references format   according  to   MDPI   journals.

Response: We check references format again and do some revisions.

Round 2

Reviewer 1 Report

This manscript could be published after minor revision of English writting.